organic chemistry/synthetic chemistry/materials science

glycoconjugates, 1,2,3-triazole, multi-component reaction, Cu–Al mixed oxide, biological activity

**Authors for correspondence:**
Leticia Lomas-Romero
e-mail: llr@xanum.uam.mx
Guillermo E. Negrón-Silva
e-mail: gns@azc.uam.mx

This article has been edited by the Royal Society of Chemistry, including the commissioning, peer review process and editorial aspects up to the point of acceptance.

# Cu–Al mixed oxide-catalysed multi-component synthesis of gluco- and allofuranose-linked 1,2,3-triazole derivatives

Ricardo Corona-Sánchez[1], Alma Sánchez-Eleuterio[2], Claudia Negrón-Lomas[3], Yarisel Ruiz Almazan[3], Leticia Lomas-Romero[1], Guillermo E. Negrón-Silva[2] and Álvaro C. Rodríguez-Sánchez[3]

[1]Departamento de Química, Universidad Autónoma Metropolitana-Iztapalapa, Av. San Rafael Atlixco 186, Leyes de Reforma 1ra Secc., 09340 Ciudad de México, Mexico
[2]Departamento de Ciencias Básicas, Universidad Autónoma Metropolitana-Azcapotzalco, Av. San Pablo No. 180, Ciudad de México C.P. 02200, Mexico
[3]Departamento de Biotecnología, Instituto Tecnológico de Estudios Superiores de Monterrey, Calle del Puente 222, Ciudad de México C.P. 14380, Mexico

RC-S, 0000-0002-2520-2798; GEN-S, 0000-0002-7886-5261

A series of carbohydrate-linked 1,2,3-triazole derivatives were synthesized in good yields from glucofuranose and allofuranose diacetonides using as key step a three-component 1,3-dipolar azide–alkyne cycloaddition catalysed by a Cu–Al mixed oxide. In this multi-component reaction, Cu–Al mixed oxide/sodium ascorbate system serves as a highly reactive, recyclable and efficient heterogeneous catalyst for the regioselective synthesis of 1,4-disubstituted 1,2,3-triazoles. The reported protocol has significant advantages over classical CuI/*N,N*-diisopropylethylamine (DIPEA) or CuSO$_4$/sodium ascorbate conditions in terms of efficiency and reduced synthetic complexity. In addition, the selective deprotection of synthesized di-*O*-isopropylidene derivatives was also carried out leading to the corresponding mono-*O*-isopropylidene products in moderate yields. Some of the synthesized triazole glycoconjugates were tested for their *in vitro* antimicrobial activity using the disc diffusion method against Gram-positive bacteria (*Staphylococcus aureus* and *Bacillus subtilis*), Gram-negative bacteria (*Escherichia coli* and *Pseudomonas aeruginosa*), as well as fungus (*Aspergillus niger*) and yeast (*Candida utilis*). The results revealed that these compounds exhibit moderate to good antimicrobial activity mainly against Gram-negative bacteria.

# 1. Introduction

1,2,3-Triazole and its derivatives are an important class of nitrogen-containing aromatic heterocyclic compounds and have become one of the most relevant topics in modern heterocyclic chemistry [1]. The 1,2,3-triazole moiety has been recognized as an attractive scaffold in drug design [2,3], due to its incorporation into many bioactive molecules as anti-cancer [4], anti-inflammatory [5], antitubercular [6], antimicrobial [7], antiviral [8] and antibacterial [9] agents. In addition to its inherent biological properties [10], 1,2,3-triazole moiety is a fascinating connecting unit which could link pharmacophores providing hybrid molecules with an increased biological profile [11]. In this context, triazolyl-glycoconjugates, molecules possessing a triazole moiety linked to a carbohydrate backbone, have emerged as valuable pharmacophores in the field of medicinal chemistry (figure 1) [12–14].

Carbohydrates are an invaluable source of stereochemically pure molecules, and they are relevant as signalling molecules and important for cellular recognition events. As a consequence, the construction of carbohydrate-based bioactive compounds has become an active research area [15]. Compounds containing the 1,2,3-triazole ring have gained increased attention in the drug-discovery field since the introduction of the 'click' chemistry concept which was reported simultaneously and independently by the groups of Meldal in Denmark [16] and Fokin and Sharpless in the United States [17,18]. Particularly, the copper(I)-catalysed azide–alkyne cycloaddition (CuAAC) to give a 1,4-disubstituted 1,2,3-triazole derivative is a powerful method for glycoconjugation due to its reliability, specificity and biocompatibility [19]. Click chemistry has great potential for use in binding between nucleic acids, lipids, proteins and carbohydrates. In this context, metabolic glycoengineering allows direct modification of living cells with substrates for click chemistry either *in vitro* or *in vivo*, becoming a powerful tool for cell transplantation and drug delivery [20]. In this sense, carbohydrates represent versatile starting substrates for the CuAAC reaction because it is possible to synthesize from them both from the azide- or the terminal alkyne-containing moieties needed for this cycloaddition reaction.

Considerable efforts have been made in order to increase the general efficiency of the process for obtaining hybrid 1,2,3-triazoles. In this aspect, heterogeneous catalysts offer several advantages over their homogeneous counterparts, such as enhanced stability, easy recovery and recycling [21,22]. For CuAAC process, almost any Cu catalyst able to furnish Cu(I) species in the reaction medium can be successfully used. In this context, the two possible approaches are either to use a Cu(I) source directly into reaction with the presence of stabilizing ligands, or to generate it either by reduction of Cu(II) salts or by oxidation of elemental Cu. *In situ* generation of Cu(I) species using Cu(II) salts such as $CuSO_4$ or copper acetate is a more effective way, since these conditions allow reactions to be carried out without an inert atmosphere and it avoids the use of anhydrous solvents, endures aqueous conditions, and at the same time maintains a high Cu(I) concentration during the course of the reaction [23].

On the other hand, hydrotalcite-like compounds constitute a class of two-dimensional materials, which are represented by a general chemical formula $[M^{II}_{1-x}M^{III}_{x}(OH)_2]^{x+}(A^{n-})_{x/n} \cdot mH_2O$, where $M^{II}$, $M^{III}$ represent divalent and trivalent metal cations, and $A^{n-}$ is an anion (usually $CO_3^{2-}$ or $NO_3^{-}$). These materials have been used mainly as catalysts for the preparation of fine chemicals, intermediates and valuable molecules [24,25]. Recently, we discovered that Cu–Al mixed oxide, generated by calcination of corresponding hydrotalcite type layered double hydroxide (LDH), in the presence of sodium ascorbate (NaAsc) as a reducing agent, efficiently catalyses the 1,3-dipolar cycloaddition of *in situ*-generated benzyl azides and phenyl acetylene in short reaction times using $EtOH/H_2O$ as a solvent [26]. In continuation with our interest in the synthesis of functional triazole derivatives [27–30], and in order to introduce the use of mixed oxides in carbohydrate CuAAC click chemistry, herein we report a rapid and facile synthesis of a series of carbohydrate-linked 1,2,3-triazole derivatives using as key step a three-component 1,3-dipolar azide–alkyne cycloaddition catalysed by Cu–Al mixed oxide, and an insight into their antimicrobial properties.

# 2. Results and discussion

The synthetic strategy adopted for the synthesis of 1,2,3-triazole–carbohydrate conjugates began with the preparation of 1,2:5,6-di-*O*-isopropylidene-*α*-D-glucofuranose **1**, which can be easily obtained through the acetonide protection of the cheap and readily available D-glucose [31]. The synthesis of 1,2:5,6-di-*O*-isopropylidene-*α*-D-allofuranose **4** involves the stereoinversion of C-3 in D-glucose diacetonide **1** through an oxidation/stereoselective reduction process as shown in scheme 1.

**Figure 1.** Some triazolyl-glycoconjugates with biological properties.

**Scheme 1.** Synthesis of *O*-propargylated glucose diacetonides.

A variety of procedures for the oxidation of **1** has been reported, including reagents such as pyridinium chlorochromate (PCC) or pyridinium dichromate (PDC) [32–34], $RuO_2$ or $RuCl_3$ and $NaIO_4$ [35,36] or oxidation procedures using activated dimethylsulfoxide (DMSO) or other oxidized forms of dimethyl sulfide, which usually involve expensive and highly toxic reagents with tedious and long reaction times of procedures [37,38]. In this work, the 1,2:5,6-di-*O*-isopropylidene-*α*-D-ribo-hexofuranos-3-ulose **3** was prepared by a modification of Okada's methodology [39], which involves the oxidation of the secondary alcohol to the corresponding ketone, by using a 2,2,6,6-tetramethyl-1-piperidinyloxy (TEMPO)-catalysed oxidation with sodium hypochlorite (NaOCl/bleach) in the presence of tetrabutylammonium hydrogensulfate (TBAHS), which was completed in only 15 min of reaction. Subsequent sodium borohydride reduction at 0°C inverts the configuration at C3-position of glucofuranose sugar to afford 1,2:5,6-di-*O*-isopropylidene-*α*-D-allofuranose **4** in 90% yield. Finally, classical propargylation reaction of diacetonides **1** and **4** using propargyl bromide and NaH as base in DMF, affords desired propargyl ethers **2** and **5** in good yields. NMR data for compounds **2** and **5** were in accordance with previous reports [40].

Then, we focused on studying the Cu–Al mixed oxide-catalysed three-component 1,3-dipolar azide–alkyne cycloaddition reaction to obtain 1,2,3-triazole–carbohydrate conjugates. Firstly, the catalyst was prepared using our previously reported method, from $Cu(NO_3)_2 \cdot 2.5H_2O$ and $AlCl_3 \cdot 6H_2O$ by co-precipitation with $Na_2CO_3$ in water [26]. After heating at 60°C for 24 h, removal of water and drying in an oven (120°C, 18 h), a green solid was obtained, which was then calcinated in a $N_2$ atmosphere (550°C, 6 h) to obtain a fine black powder material corresponding to Cu–Al mixed oxide. The structure of mixed oxide was confirmed by X-ray diffraction. Figure 2 shows the diffractogram of calcinated layered double hydroxide (HLD), with the characteristic pattern of a periclase-like structure with (110), (111), (202), (022), (113), (311) and (220) plane reflections, typical of CuO, consistent with what was previously described.

With the raw materials and the catalyst in hand, in our initial experiment, the multi-component reaction between the glucose diacetonide **2**, benzyl chloride and sodium azide was conducted in the presence of 10 mg of the Cu–Al mixed oxide and the same amount of sodium ascorbate at 80°C; with conventional heating for 18 h in ethanol–water as solvent (table 1, entry 1).

Under these conditions, the 1,2,3-triazole–carbohydrate conjugate **7a** was obtained as a white crystalline solid in 69% yield after purification by crystallization with dichloromethane–hexane. Thus, we have demonstrated that Cu–Al mixed oxide can catalyse this three-component 1,3-dipolar azide–alkyne cycloaddition reaction in a regioselective manner, giving exclusively the 1,4-disubstituted

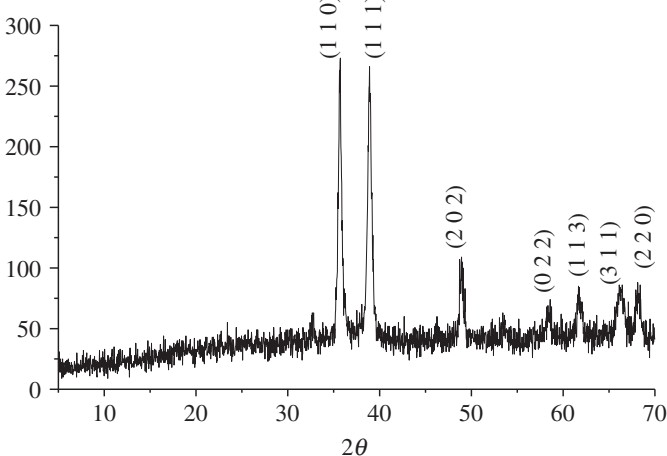

**Figure 2.** X-ray diffraction patterns of the Cu(Al)O mixed oxide.

**Table 1.** Optimization of reaction conditions for three-component 1,3-dipolar azide–alkyne cycloaddition catalysed by Cu–Al mixed oxide.[a]

| entry | catalyst | solvent | heating | time | yield (%)[b] |
|---|---|---|---|---|---|
| 1[c] | Cu–Al mixed oxide | EtOH–$H_2O$ | conventional | 18 h | 69 |
| 2[c] | Cu–Al mixed oxide | EtOH–$H_2O$ | microwave | 10 min | 64 |
| 3 | Cu–Al mixed oxide | EtOH–$H_2O$ | microwave | 10 min | 81 |
| 4 | Cu–Al mixed oxide | EtOH–$H_2O$ | microwave | 30 min | 83 |
| 5 | Cu–Al mixed oxide | DMF | microwave | 10 min | 52 |
| 6 | Cu–Al mixed oxide | MeCN | microwave | 10 min | 45 |
| 7[d] | $CuSO_4 \cdot 5H_2O$/NaAsc | EtOH–$H_2O$ | microwave | 30 min | ND |
| 8[e] | CuI/DIPEA | EtOH–$H_2O$ | microwave | 10 min | 83 |

[a]Reaction conditions: alkyne **2** (1 mmol), benzyl chloride (1.2 mmol), $NaN_3$ (1.2 mmol), Cu–Al mixed oxide (30 mg), sodium ascorbate (30 mg) and EtOH:$H_2O$ (2 ml 3 : 1) at 80°C.
[b]Yields of the isolated product after purification by crystallization.
[c]The reaction was performed in the presence of 10 mg of catalyst and 10 mg of sodium ascorbate.
[d]The reaction was performed in the presence of $CuSO_4 \cdot 5H_2O$ (20 mol%) and sodium ascorbate (20 mol%).
[e]The reaction was performed in the presence of CuI (20 mol%) and DIPEA (30 mol%). ND, not detected.

triazole. To determine the optimal conditions for the reaction, a series of experiments were performed varying parameters such as catalyst loading, solvent and heating method (table 1, entries 2–7). The effectiveness of the microwave irradiation and the conventional heating for this three-component click reaction was compared. In this case, a slightly lower yield than that obtained with the conventional heating was found (table 1, entry 2). However, although both methods are efficient, the microwave-assisted reaction was faster (10 min) in comparison with conventional thermal conditions, which require 18 h to complete the reaction. We also found that 30 mg of catalyst provides the best result, with optimum yields of the glycoconjugate product (83%), using an EtOH:$H_2O$ (3 : 1) mixture (table 1, entry 3); whereas under the same conditions, other solvents gave lower yields (table 1, entries 5 and 6). For comparative purposes, homogeneous processes using the typical conditions employed in carbohydrate CuAAC click chemistry namely, $CuSO_4 \cdot 5H_2O$/sodium ascorbate or CuI/*N*,*N*-

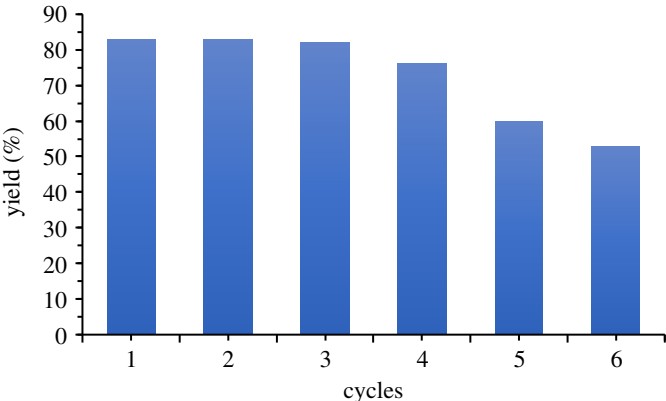

**Scheme 2.** Two-component 1,3-dipolar azide–alkyne cycloaddition catalysed by Cu–Al mixed oxide.

**Figure 3.** Recyclability of catalyst for the three-component 1,3-dipolar azide–alkyne cycloaddition.

diisopropylethylamine (DIPEA) systems were performed (table 1, entries 5 and 6). The Cu(I) salt and DIPEA were used in the reaction, under the same solvent and heating conditions, yielding 83% of triazole **7a** after column chromatography. Surprisingly, the $CuSO_4 \cdot 5H_2O$/sodium ascorbate system using microwave irradiation with $EtOH/H_2O$ as solvent was not efficient for this transformation since 1,2,3-triazole compound could not be detected and only the formation of benzyl azide took place. This is probably due to the fact that Cu(I) species formed by the action of a reducing agent are not stable enough under the used conditions and an external stabilizing ligand needs to be added, which becomes a drawback for this method. Thus, we demonstrate that the described method is efficient and does not need any additional stabilizing ligands. Finally, although our ultimate goal was to obtain a heterogeneous and recyclable catalyst for the multi-component version of azide–alkyne cycloaddition applicable in carbohydrate CuAAC click chemistry, we also performed an experiment with the standard two-component reaction between phenylacetylene and benzyl azide catalysed by Cu–Al mixed oxide. It was expected that the reaction with the preformed organic azide was faster than with the multi-component approach; however, after 30 min using microwave (MW) irradiation at 80°C in $EtOH/H_2O$ as solvent, only 43% yield of 1,4-disubstituted 1,2,3-triazole **7a** was obtained, besides the recovery of part of the benzyl azide used as starting material (scheme 2). This could be attributed to the low solubility of benzyl azide in $EtOH/H_2O$ system and demonstrates that the three-component 1,3-dipolar azide–alkyne cycloaddition reaction catalysed by Cu–Al mixed oxide remains the best approach to perform this reaction.

It is worth noting that, under optimized conditions, the mixed oxide catalyst was successfully recovered, reactivated by calcination at 500°C, and re-used twice more in the same click reaction with similar efficiency. Subsequent to this reuse, the materials probably lose their initial structure and therefore their catalytic activity (figure 3).

Encouraged by these results, once the reaction was optimized and the catalyst recyclability was evaluated, we synthesized two series of 1,2,3-triazole–carbohydrate conjugates using the *O*-propargylated gluco- or allofuranose derivatives with a range of different *p*-halobenzyl chlorides in order to obtain 1,2,3-triazole–carbohydrate conjugates which could be used as valuable substrates for cross-coupling reactions (table 2).

Gratifyingly, in nearly all cases, very good yields of triazoles **7a–e** and **8a–e** were obtained and halogen substituents at the substrates were well tolerated. Both diastereomeric series of carbohydrate–triazole conjugates were fully characterized by FT-IR, HRMS, [1]H- and [13]C-NMR spectroscopy. The IR spectra of compounds **7a–e** and **8a–e** showed the characteristic absorption assigned to the N=N stretching for triazole compounds that appear nearly around 1490 cm$^{-1}$ and three or four intense absorption bands

**Table 2.** Synthesis of carbohydrate-linked 1,2,3-triazole derivatives via three-component 1,3-dipolar azide–alkyne cycloaddition catalysed by Cu–Al mixed oxide.[a]

**2 or 5** + **6a–d**

Cu/Al mixed oxide
sodium ascorbate
EtOH/$H_2O$
MW, 80°C
10 min

**7a–e** (glucofuranose series)
**8a–e** (allofuranose series)

| entry | diacetonide | benzyl chloride | R | product | yield (%)[b] |
|---|---|---|---|---|---|
| 1 | | **6a** | —H | **7a** | 81 |
| 2 | | **6b** | —F | **7b** | 67 |
| 3 | **2** | **6c** | —Cl | **7c** | 73 |
| 4 | | **6d** | —Br | **7d** | 58 |
| 5 | | **6e** | —I | **7e** | 53 |
| 6 | | **6a** | —H | **8a** | 89 |
| 7 | | **6b** | —F | **8b** | 79 |
| 8 | **5** | **6c** | —Cl | **8c** | 78 |
| 9 | | **6d** | —Br | **8d** | 89 |
| 10 | | **6e** | —I | **8e** | 53 |

[a]Reaction conditions: Alkyne **2** or **5** (1 mmol), benzyl chloride (1.2 mmol), $NaN_3$ (1.2 mmol), Cu–Al mixed oxide (30 mg), sodium ascorbate (30 mg) and EtOH : $H_2O$ (2 ml 3 : 1) at 80℃.
[b]Yields of the isolated product after purification.

around 1025–1130 cm$^{-1}$ assigned to typical C–O stretching of aliphatic ethers of carbohydrate fragment. Furthermore, the identity of the carbohydrate–triazole conjugates presented in this work was supported by high-resolution mass spectrometry (HRMS) which confirmed the expected mass for all compounds in accordance with their molecular formula. The analysis of the NMR spectroscopic data ($^1$H, $^{13}$C, homonuclear correlation spectroscopy (COSY), heteronuclear single quantum correlation (HSQC) and heteronuclear multiple bond correlation (HMBC) experiments) reveals the diagnostic singlet in the resonance of the $^1$H NMR due to triazolyl protons in the region of 7.50 ppm and around 122.0 ppm in $^{13}$C NMR due to C-5 of the triazole ring, which confirms the formation of glycoconjugates. Representative $^1$H NMR spectra of diastereomeric compounds **7c** and **8c** are shown in figure 4.

An analysis of spectroscopic data for both diastereomeric carbohydrate–triazole derivatives revealed a diagnostic pattern associated with the methylene protons (H-25). For glucofuranose derivative **7c**, a simple signal at 5.48 ppm is assigned to H-25; while for analogue allofuranose derivative **8c**, the same methylene group appearing as an AB quartet system in 5.49 ppm ($J = 12.7$ Hz). Besides, the double of doublet at 3.97 ppm ($J = 8.7$, 5.5 Hz) assigned to H-5 in **7c** is slightly downshifted to 4.06 ppm ($J = 8.7$, 3.4 Hz) in the case of allofuranose derivative **8c**. The regioselectivity of the cycloaddition reaction was elucidated by using $^1$H-$^{13}$C long-range NMR experiments (HMBC). The selective formation of 1,4-disubstituted 1,2,3-triazoles was confirmed by a long-range correlation between the methylene protons H-19 and H-25 with the carbon-bearing H-21 in the triazole ring, since in 1,5-regioisomer the correlation between the benzyl protons and the carbon bearing the proton in the triazole ring would not be present (figure 5).

On the other hand, one of the advantages of di-O-isopropylidene gluco- and allofuranose derivatives is that these compounds could be selectively deprotected to generate diols that can be subsequently functionalized for their use in different fields of scientific research. Tiwari and co-workers [41] reported the synthesis of a series of glycosyl triazoles and their evaluation as efficient ligands in C-heteroatom coupling reactions by using a mono-O-isopropylidene glucofuranose derivative as starting material. Srinivas et al. [42,43] reported a 1,2,3-triazole-linked allofuranose compound that can be a key intermediary in the preparation of a series of heterocyclic-containing carbohydrate–triazole hybrids with potential antimicrobial activity. The synthesis of such compounds involves a selective acidic hydrolysis of the corresponding acetonide to yield a chiral diol. Besides, it is being observed

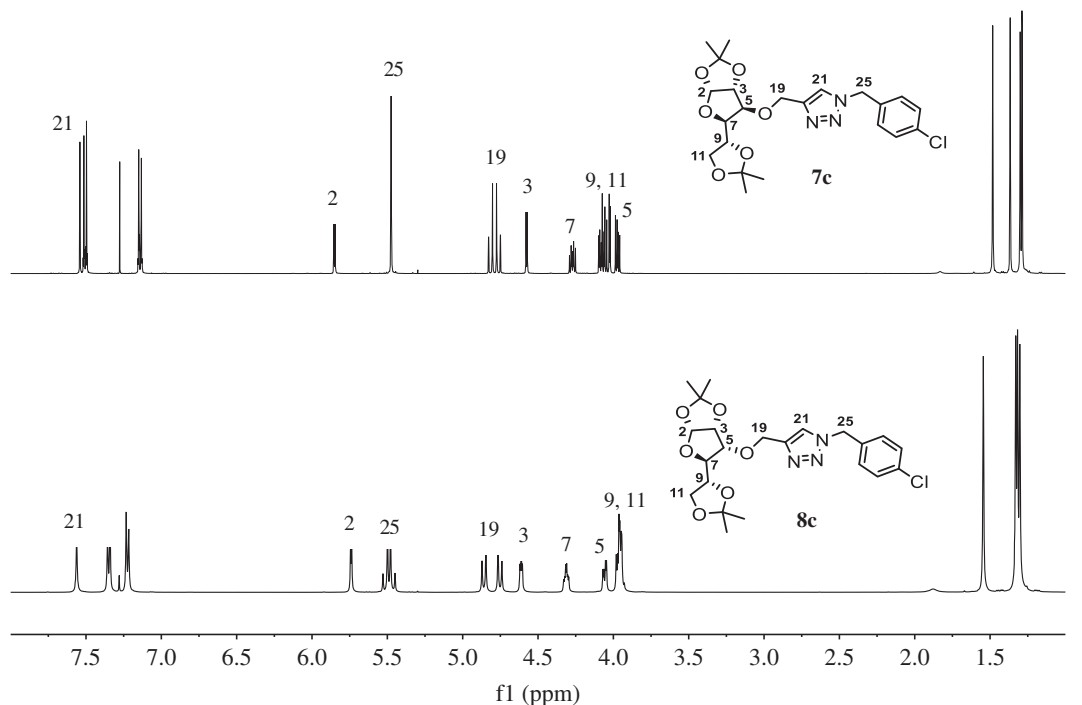

**Figure 4.** $^1$H NMR spectra of carbohydrate–triazole conjugates **7c** and **8c**.

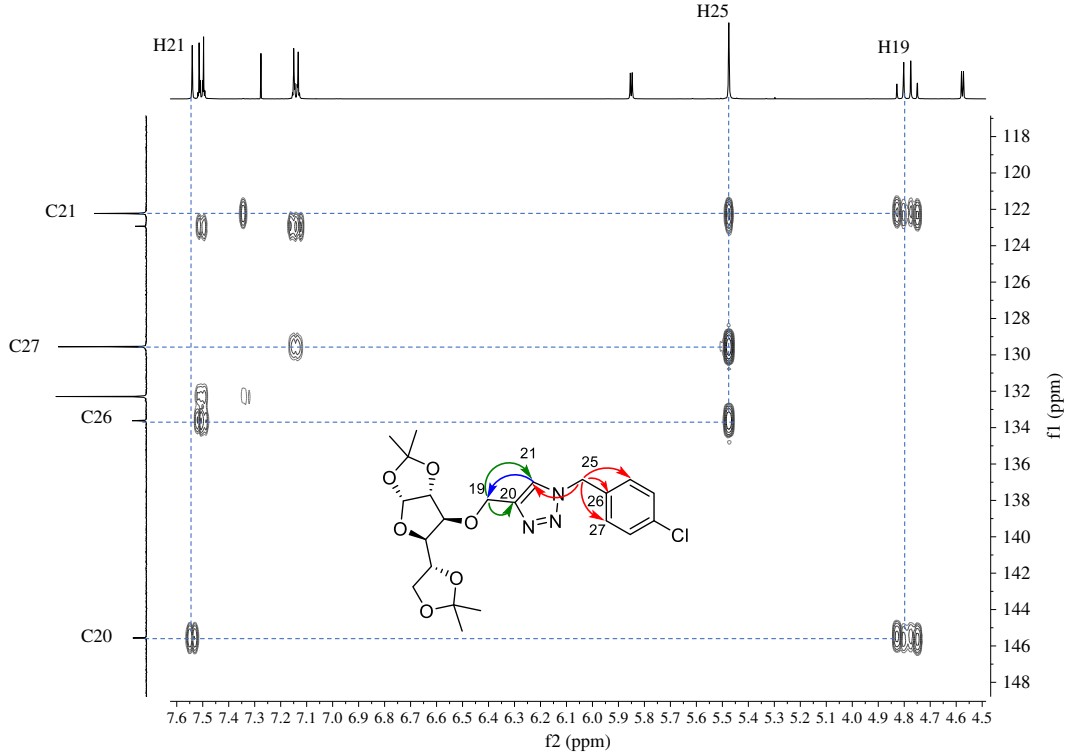

**Figure 5.** Expansion of $^1$H-$^{13}$C HMBC spectrum highlighting the correlations of triazole ring in compound **7c**.

that the presence of multiple free hydroxyl groups in this type of structural scaffolding (unprotected carbohydrate–triazole hybrids) increases not only the solubility in the biological systems favoured by hydrogen bonds but also enhances the affinity and selectivity towards recognition proteins of carbohydrates [44]. In order to obtain new deprotected carbohydrate-linked 1,2,3-triazole, and considering the potential application of these products, we proceeded the removal of the acetonide protecting groups of carbohydrate–triazole conjugates **7a–e** and **8a–e** under acidic conditions. We

**Table 3.** Synthesis of carbohydrate-linked 1,2,3-triazole derivatives via three-component 1,3-dipolar azide–alkyne cycloaddition catalysed by Cu–Al mixed oxide.[a]

7a–e (glucofuranose series)
8a–e (allofuranose series)

9a–e and 10a–e

| entry | triazole–carbohydrate | | R | product | yield (%)[b] |
|---|---|---|---|---|---|
| 1 | | **7a** | —H | **9a** | 45 |
| 2 | | **7b** | —F | **9b** | 46 |
| 3 | **9** | **7c** | —Cl | **9c** | 42 |
| 4 | | **7d** | —Br | **9d** | 50 |
| 5 | | **7e** | —I | **9e** | 40 |
| 6 | | **8a** | —H | **10a** | 47 |
| 7 | | **8b** | —F | **10b** | 38 |
| 8 | **10** | **8c** | —Cl | **10c** | 43 |
| 9 | | **8d** | —Br | **10d** | 42 |
| 10 | | **8e** | —I | **10e** | 48 |

[a]Reaction conditions: carbohydrate–triazole conjugate **7a**–**e** or **8a**–**e** (50 mg, 0.11 mmol) and HCl : MeOH (2 ml 4 N: 4 ml), 5 h at 25℃.
[b]Yields of the isolated product after purification.

**Table 4.** *In vitro* antimicrobial activity data of compounds **7a**–**e**.

| compound | zone of inhibition (mm) | | | | | |
|---|---|---|---|---|---|---|
| | Gram-positive bacteria | | Gram-negative bacteria | | fungi | |
| | *B. subtilis* | *S. aureus* | *E. coli* | *P. aeruginosa* | *C. utilis* | *A. niger* |
| **7a** | 0.6 | 1.3 | 1.3 | 2.6 | 0.7 | 0 |
| **7b** | 0.6 | 0.6 | 2 | 2 | 0.3 | 0 |
| **7c** | 0 | 1 | 1.6 | 1.3 | 0.7 | 0 |
| **7d** | 0.3 | 1 | 2.3 | 2 | 0.7 | 0 |
| **7e** | 0 | 1 | 2.3 | 2 | 1.3 | 0 |
| streptomycin | — | — | 6 | 3 | — | — |
| amphotericin | — | — | — | — | 8 | 3 |
| penicillin | 10 | 10 | — | — | — | — |

found that HCl–MeOH system at room temperature (RT) was a clean method to obtain the desired deprotected derivatives **9a–e** and their diastereomeric series of derivatives **10a–e** in moderate yields. These results are summarized in table 3. According to the ¹H NMR spectra, the absence of two methyl groups around 1.20–1.40 ppm confirms the deprotection in the 5,6 positions of carbohydrate–triazole **7a–e** and **8a–e**. The hydrolysis products were easily purified by column chromatography, allowing access to two new families of carbohydrate–triazole derivatives **9a–e** and **10a–e** through a simple structural modification in green and low-cost conditions.

Finally, in order to have an insight into the possible biological activity of synthesized compounds, preliminary studies about the antimicrobial activity of triazoles **7a–e** against different Gram-positive bacteria (*Staphylococcus aureus* and *Bacillus subtilis*), Gram-negative bacteria (*Escherichia coli* and *Pseudomonas aeruginosa*), as well as fungus (*Aspergillus niger*) and yeast (*Candida utilis*), were determined by using the disc diffusion test (table 4).

The results showed that almost all tested compounds exhibit a moderate antimicrobial activity against the tested bacteria strains. In general, compounds **7a–e** were more active against Gram-negative than for Gram-positive bacteria, where compound **7a** was the most active compound against *P. aeruginosa* with an inhibition similar to that of streptomycin used as reference. No evident correlation between the electronic nature of halogen substituent and antibacterial activity was found. In the case of the antifungal activity, the tested triazole exhibited low activity against *C. utilis* and no antifungal activity was observed against *A. niger* strain.

# 3. Conclusion

We developed an easy and efficient method for the synthesis of two diastereomeric series of 1,2,3-triazole–carbohydrate conjugates derived from gluco- and allofuranose diacetonides through a three-component 1,3-dipolar azide–alkyne cycloaddition catalysed by Cu–Al mixed oxide. In this multi-component reaction, the microwave irradiation dramatically reduces the reaction time and the catalyst could be recycled and re-used at least for two runs without notable reduction of its catalytic activity. Some advantages of this protocol are mild reaction conditions, good yields, short reaction times and easy work-up and isolation. Besides, both protected and deprotected triazole–carbohydrate conjugates that were synthesized could be suitable starting materials for preparing novel compounds with an interesting biological profile. Further research to extend the substrate scope is under way in our laboratory.

# 4. Material and methods

All reagents were obtained from commercial suppliers and used without further purification. Merck silica gel (type 60, 0.063–0.200 mm) was used for column chromatography. All compounds were characterized by IR spectra, recorded on an FT-IR Bruker Tensor 27 spectrophotometer, by means of attenuated total reflection (ATR) technique, and all data is expressed as wavenumbers ($cm^{-1}$). Melting points were obtained on a Fisher–Johns apparatus and are uncorrected. NMR spectra were recorded on a Bruker Ascend-400 (400 MHz) and Bruker Avance DMX-500 (500 MHz) spectrometers in $CDCl_3$ or acetone-d6 and chemical shifts are given in ppm with tetramethylsilane (TMS) as the reference. Direct analysis in real time mass spectrometry (DART-MS) were obtained on a Jeol AccuTOF; the values of the signals are expressed as mass/charge units ($m/z$). Microwave irradiation experiments were performed using a Discover System (CEM Corporation, Matthews, NC, USA) single-mode microwave with standard sealed microwave glass vials. The reaction temperature was monitored by an IR sensor on the outside wall of the reaction.

## 4.1. Synthesis of 1,2:5,6-di-*O*-isopropylidene-$\alpha$-D-glucofuranose (**1**)

A solution of D-glucose (5 g, 27.78 mmol) in dry acetone (250 ml) was added concentrated sulfuric acid (1.2 ml) at room temperature. The reaction mixture was stirred vigorously at room temperature for 6 h. Anhydrous copper(II) sulfate (15 g) was added to the reaction mixture and stirring was continued at room temperature for 18 h. The reaction mixture was then neutralized with sodium bicarbonate and the inorganic materials were filtered off. The filtrate was evaporated under reduced pressure to leave a white solid, which was partitioned between $CH_2Cl_2$ and $H_2O$. The organic layer was dried over $Na_2SO_4$, filtered and evaporated under reduced pressure providing the title compound as a white solid. Recrystallization from hexane gave the pure diacetal.

## 4.2. Synthesis of 1,2:5,6-di-*O*-isopropylidene-$\alpha$-D-allofuranose

The D-glucofuranose **1** (1 g, 3.84 mmol) was dissolved in a mixture of $CH_2Cl_2$/NaClO (4 : 4 ml) then was added TBAHS (0.260 g, 0.768 mmol) and TEMPO (0.090 g, 0.576 mmol). The reaction mixture was stirred vigorously at 35°C for 15 min. After this time, the reaction mixture was extracted three times with $CH_2Cl_2$ (30 ml). The organic phase was dried with $Na_2SO_4$ and evaporated under reduced pressure, and the resultant residue was dissolved in 10 ml of methanol. The reaction mixture was stirred for 10 min at 0°C before the addition of $NaBH_4$ (0.290 g, 7.68 mmol) and allowed to react for 40 min at 25°C. After this time, the reaction was quenched with 30 ml of water, and the aqueous layer was extracted three times with ethyl acetate, was dried over anhydrous $Na_2SO_4$, filtered, concentrated under vacuum and purified by column chromatography (ethyl acetate : hexane 2 : 1) afforded desired sugar **4** [45] in 90%, $[\alpha]_D$ + 34.3 (c 1, $CHCl_3$), m.p. 75–78°C .

## 4.3. Synthesis of propargyl carbohydrate esters 2 and 5

A solution of 1,2:5,6-di-O-isopropylidene-α-D-glucofuranose 2 or 1,2:5,6-di-O-isopropylidene-α-D-allofuranose 5 (1 g, 3.84 mmol) in anhydrous DMF (10 ml) was cooled to 0°C and sodium hydride 60% dispersion in mineral oil (320 mg, 9.0 mmol) was added. The reaction mixture was stirred at 0°C under nitrogen atmosphere for 30 min. Propargyl bromide 80 wt% in toluene (0.47 ml, 1.2 mmol) was added at 0°C and allowed to stir for 12 h at room temperature. Upon completion of the reaction, remaining sodium hydride quenched by water, the solvent was removed under reduced pressure and extracted with ethyl acetate (3 × 15 ml). The organic layer was dried over anhydrous $Na_2SO_4$, filtered, concentrated under vacuum and purified by column chromatography (ethyl acetate : hexane 85 : 15) afforded desired sugar-based alkyne.

## 4.4. General procedure for the synthesis of carbohydrate-1,2,3-triazole conjugates

A mixture of catalyst (30 mg) and EtOH–$H_2O$ (2 ml, 3 : 1 v/v) was placed in a microwave tube having a magnetic stirrer. Subsequently, alkyne 2 or 5 (298 mg, 1 mmol), benzyl chloride 6a–e (1.2 mmol), $NaN_3$ (78 mg, 1.2 mmol) and sodium ascorbate (30 mg) were added to the mixture, which was heated under microwave irradiation (30 W, 80°C) for 10 min. Then, the reaction mixture was diluted with dichloromethane and filtered through a short plug of silica. The organic extract was dried with $Na_2SO_4$ and the solvent was evaporated giving the corresponding 1,2,3-triazole, which was then purified by recrystallization from dichloromethane–hexanes.

### 4.4.1. 1,2:5,6-Di-O-isopropylidene-3-O-(1-benzyl-1H-1,2,3-triazol-4-yl-methyl)-α-D-glucofuranose (7a)

White solid, 349 mg (81%), m.p. = 125–127°C; $[\alpha]_D$ -37.0 (c 1, CHCl₃). ¹H NMR (500 MHz, CDCl₃) δ 7.54 (s, 1H, H-21), 7.37 (d, J = 6.8 Hz, 3H, ArH), 7.31–7.23 (m, 2H, ArH), 5.85 (d, J = 3.7 Hz, 1H, H-2), 5.52 (s, 2H, H-25), 4.81, 476 (ABq, J = 12.7 Hz, 2H, H-19), 4.57 (d, J = 3.8 Hz, 1H, H-3), 4.27 (dt, J = 8.2, 5.8 Hz, 1H, H-7), 4.15–4.01 (m, 3H, H-9 and H-11), 3.96 (dd, J = 8.6, 5.5 Hz, 1H, H-5), 1.48 (s, 3H, –CH₃), 1.36 (s, 3H, –CH₃), 1.30 (s, 3H, –CH₃), 1.28 (s, 3H, –CH₃). ¹³C NMR (126 MHz, CDCl₃) δ 145.4, 134.7, 129.2, 128.8, 128.0, 122.3, 111.9, 109.0, 105.3, 82.6, 81.8, 81.1, 72.4, 67.4, 64.2, 54.2, 26.8 (2C), 26.2, 25.4. FT-IR/ATR $v_{max}$ cm⁻¹: 2931, 1452, 1370, 1256, 1213, 1164, 1129, 1074, 1025, 843. HRMS (ESI-TOF) calculated for $C_{22}H_{30}N_3O_6$ [M + H]⁺ 432.21346, found 432.22270.

### 4.4.2. 1,2:5,6-Di-O-isopropylidene-3-O-(1-(4-fluorobenzyl)-1H-1,2,3-triazol-4-yl-methyl)-α-D-glucofuranose (7b)

White solid, 279 mg (62%), m.p. = 127–129°C; $[\alpha]_D$ -29.0 (c 1, CHCl₃). ¹H NMR (500 MHz, CDCl₃) δ 7.54 (s, 1H, H-21), 7.34–7.23 (m, 2H, ArH), 7.12–7.02 (m, 2H, ArH), 5.85 (d, J = 3.6 Hz, H-2), 5.49 (s, 2H, H-25), 4.81, 476 (ABX, J = 12.6, 0.7, 2H, H-19), 4.58 (d, J = 3.7 Hz, 1H, H-3), 4.27 (ddd, J = 8.1, 6.1, 5.5 Hz, 1H, H-7), 4.10–4.01 (m, 3H, H-9 and H-11), 3.97 (dd, J = 8.6, 5.5 Hz, 1H, H-5), 1.48 (s, 3H, –CH₃), 1.37 (s, 3H, –CH₃), 1.30 (s, 3H, –CH₃), 1.29 (s, 3H, –CH₃). ¹³C NMR (126 MHz, CDCl₃) δ 163.9, 161.9, 145.5, 130.5, 129.9, 129.8, 122.2, 116.2, 116.1, 111.9, 109.0, 105.2, 82.6, 81.8, 81.1, 72.4, 67.4, 64.2, 53.4, 26.8, 26.8, 26.2, 25.4. FT-IR/ATR $v_{max}$ cm⁻¹: 2985, 1511, 1371, 1259, 1215, 1158, 1132, 1027, 1023, 844. HRMS (ESI-TOF) calculated for $C_{22}H_{29}FN_3O_6$ [M + H]⁺ 450.20404, found 450.20056.

### 4.4.3. 1,2:5,6-Di-O-isopropylidene-3-O-(1-(4-chlorobenzyl)-1H-1,2,3-triazol-4-yl-methyl)-α-D-glucofuranose (7c)

White solid, 341 mg (73%) m.p. = 113–114°C; $[\alpha]_D$ -31.0 (c 1, CHCl₃). ¹H NMR (500 MHz, CDCl₃) δ 7.54 (s, 1H, H-21), 7.51 (d, J = 8.6 Hz, 2H, ArH), 7.14 (d, J = 8.7 Hz, 2H, ArH), 5.85 (d, J = 3.7 Hz, 1H, H-2), 5.48 (s, 2H, H-25), 4.82, 4.76 (ABX, J = 12.6, 0.7, 2H, H-19), 4.58 (d, J = 3.7 Hz, 1H, H-3), 4.27 (ddd, J = 8.2, 6.2, 5.5 Hz, 1H, H-7), 4.12–4.00 (m, 3H, H-9 and H-11), 3.97 (dd, J = 8.7, 5.5 Hz, 1H, H-5), 1.48 (s, 3H, –CH₃), 1.37 (s, 3H, –CH₃), 1.30 (s, 3H, –CH₃), 1.29 (s, 3H, –CH₃). ¹³C NMR (126 MHz, CDCl₃) δ 145.6, 133.6, 132.3, 129.6, 122.9, 122.2, 111.8, 109.0, 105.2, 82.6, 81.8, 81.1, 77.3, 77.0, 76.8, 72.3, 67.4, 64.1, 53.5, 26.8, 26.8, 26.2, 25.4. IR/ATR $v_{max}$ cm⁻¹: 2979, 1494, 1371, 1257, 1214, 1129, 1085, 1071, 1024, 846. HRMS (ESI-TOF) calculated for $C_{22}H_{29}ClN_3O_6$ [M + H]⁺ 468.17154, found: 468.17553.

### 4.4.4. 1,2:5,6-Di-O-isopropylidene-3-O-(1-(4-bromobenzyl)-1H-1,2,3-triazol-4-yl-methyl)-α-D-glucofuranose (7d)

White solid, 295 mg (58%), m.p. = 115–116°C; $[\alpha]_D$ -26.0 (c 1, CHCl₃). ¹H NMR (500 MHz, CDCl₃) δ 7.54 (s, 1H, H-21), 7.35 (d, J = 8.7 Hz, 2H, ArH), 7.20 (d, J = 8.7 Hz, 2H, ArH), 5.85 (d, J = 3.7 Hz, 1H, H-2), 5.49

(s, 2H, H-25), 4.82, 476 (ABX, $J$ = 12.6, 0.7, 2H, H-19), 4.58 (d, $J$ = 3.7 Hz, 1H, H-3), 4.27 (ddd, $J$ = 8.2, 6.2, 5.5 Hz, 1H, H-7), 4.16–4.01 (m, 3H, H-9 and H-11), 3.97 (dd, $J$ = 8.6, 5.5 Hz, 1H, H-5), 1.48 (s, 3H, –CH$_3$), 1.37 (s, 3H, –CH$_3$), 1.30 (s, 3H, –CH$_3$), 1.29 (s, 3H, –CH$_3$). $^{13}$C NMR (126 MHz, CDCl$_3$) $\delta$ 145.6, 133.6, 132.3, 129.6, 122.9, 122.2, 111.8, 109.0, 105.2, 82.6, 81.8, 81.1, 77.2, 72.3, 67.4, 64.1, 53.5, 26.8, 26.8, 26.2, 25.4. FT-IR/ATR $v_{max}$ cm$^{-1}$: 2981, 1490, 1370, 1259, 1215, 1124, 1052, 1072, 1014, 845, 779. HRMS (ESI-TOF) calculated for C$_{22}$H$_{29}$BrN$_3$O$_6$ [M + H]$^+$ 510.12397, found 510.12319.

### 4.4.5. 1,2:5,6-Di-*O*-isopropylidene-3-*O*-(1-(4-iodobenzyl)-1H-1,2,3-triazol-4-yl-methyl)-$\alpha$-D-glucofuranose (**7e**)

White solid, 295 mg (53%), m.p. = 108–109°C; [$\alpha$]$_D$ -53.0 (c 1, CHCl$_3$). $^1$H NMR (500 MHz, CDCl$_3$) $\delta$ 7.71 (d, $J$ = 8.5 Hz, 2H, ArH), 7.54 (s, 1H, H-21), 7.01 (d, $J$ = 8.5 Hz, 2H, ArH), 5.85 (d, $J$ = 3.7 Hz, 1H, H-2), 5.46 (s, 2H, H-25), 4.81, 4.76 (ABX, $J$ = 12.7, 0.7 Hz, 2H, H-19), 4.57 (d, $J$ = 3.8, 1H, H-3), 4.27 (ddd, $J$ = 8.2, 6.2, 5.5 Hz, 1H, H-7), 4.13–4.01 (m, 3H, H-9 and H-11), 3.97 (dd, $J$ = 8.6, 5.5 Hz, 1H, H-5), 1.48 (s, 3H, –CH$_3$), 1.37 (s, 3H, –CH$_3$), 1.30 (s, 3H, –CH$_3$), 1.29 (s, 3H, –CH$_3$). $^{13}$C NMR (126 MHz, CDCl$_3$) $\delta$ 145.6, 138.3, 134.3, 129.7, 129.7, 122.2, 111.8, 109.0, 105.2, 94.5, 82.59, 81.8, 81.1, 77.2, 72.3, 67.4, 64.1, 53.5, 26.8, 26.8, 26.2, 25.4. FT-IR/ATR $v_{max}$ cm$^{-1}$: 2987, 2343, 1486, 1372, 1214, 1075, 1052, 1008, 846, 783. HRMS (ESI-TOF) calculated for C$_{22}$H$_{29}$IN$_3$O$_6$ [M + H]$^+$ 558.11010, found 558.06591.

### 4.4.6. 1,2:5,6-Di-*O*-isopropylidene-3-*O*-(1-benzyl-1H-1,2,3-triazol-4-yl-methyl)-$\alpha$-D-allofuranose (**8a**)

White solid, 383 mg (89%), m.p. = 107–108°C; [$\alpha$]$_D$ + 63.0 (c 1, CHCl$_3$). $^1$H NMR (500 MHz, CDCl$_3$) $\delta$ 7.56 (s, 1H, H-21), 7.46–7.33 (m, 3H, ArH), 7.32–7.22 (m, 2H, ArH), 5.73 (d, $J$ = 3.7 Hz, 1H, H-2), 5.54, 5.49 (ABq, $J$ = 14.8 Hz, 2H, H-25), 4.85, 4.75 (ABX, $J$ = 12.8, 0.6 Hz, 2H, H-19), 4.61–4.58 (m, 1H, H-3), 4.30 (td, $J$ = 6.9, 3.4 Hz, 1H, H-7), 4.05 (dd, $J$ = 8.7, 3.4 Hz, 1H, H-5), 4.00–3.90 (m, 3H, H-9 and H-11), 1.54 (s, 3H, –CH$_3$), 1.32 (s, 3H, –CH$_3$), 1.32 (s, 3H, –CH$_3$), 1.29 (s, 3H, –CH$_3$). $^{13}$C NMR (126 MHz, CDCl$_3$) $\delta$ 145.0, 134.5, 129.1, 128.7, 128.1, 122.9, 112.8, 109.5, 103.8, 77.9, 77.8, 77.2, 74.8, 65.0, 63.6, 54.1, 26.7, 26.4, 26.0, 25.00. FT-IR/ATR $v_{max}$ cm$^{-1}$: 2927, 1457, 1372, 1266, 1205, 1163, 1115, 1108, 871, 723. HRMS (ESI-TOF) calculated for C$_{22}$H$_{30}$N$_3$O$_6$ [M + H]$^+$ 432.21346, found 432.22256.

### 4.4.7. 1,2:5,6-Di-*O*-isopropylidene-3-*O*-(1-(4-fluorobenzyl)-1H-1,2,3-triazol-4-yl-methyl)-$\alpha$-D-allofuranose (**8b**)

White solid, 354 mg (79%), m.p. = 121–123°C; [$\alpha$]$_D$ + 80.0 (c 1, CHCl$_3$). $^1$H NMR (500 MHz, CDCl$_3$) $\delta$ 7.56 (s, 1H, H-21), 7.33–7.24 (m, 2H, ArH), 7.06 (t, $J$ = 8.6 Hz, 2H, ArH), 5.74 (d, $J$ = 3.7 Hz, 1H, H-2), 5.52, 5.46 (ABq, $J$ = 14.8 Hz, 2H, H-25), 4.86, 4.75 (ABq, $J$ = 12.8 Hz, 2H, H-19), 4.61 (t, $J$ = 4.1 Hz, 1H, H-3), 4.31 (td, $J$ = 6.9, 3.4 Hz, 1H, H-7), 4.06 (dd, $J$ = 8.7, 3.4 Hz, 1H, H-5), 4.02–3.89 (m, 3H, H-9 and H-11), 1.54 (s, 3H, –CH$_3$), 1.33 (s, 3H, –CH$_3$), 1.32 (s, 3H, –CH$_3$), 1.30 (s, 3H, –CH$_3$). $^{13}$C NMR (126 MHz, CDCl$_3$) $\delta$ 163.8, 161.9, 145.2, 130.4, 130.3, 130.0, 129.9, 122.8, 116.2, 116.0, 112.8, 109.6, 103.8, 79.0, 78.0, 77.8, 74.77, 65.0, 63.6, 53.4, 26.7, 26.4, 26.0, 24.9. FT-IR/ATR $v_{max}$ cm$^{-1}$: 2988, 1508, 1371, 1220, 1163, 1117, 1085, 1032, 850, 783. HRMS (ESI-TOF) calculated for C$_{22}$H$_{29}$FN$_3$O$_6$ [M + H]$^+$ 450.20404, found 450.20824.

### 4.4.8. 1,2:5,6-Di-*O*-isopropylidene-3-*O*-(1-(4-chlorobenzyl)-1H-1,2,3-triazol-4-yl-methyl)-$\alpha$-D-allofuranose (**8c**)

White solid, 364 mg (78%), m.p. = 110–111°C; [$\alpha$]$_D$ + 50.0 (c 1, CHCl$_3$). $^1$H NMR (500 MHz, CDCl$_3$) $\delta$ 7.56 (s, 1H, H-21), 7.35 (d, $J$ = 8.4 Hz, 2H, ArH), 7.22 (d, $J$ = 8.2 Hz, 2H, ArH), 5.74 (d, $J$ = 3.7 Hz, 1H, H-2), 5.51, 5.46 (ABq, $J$ = 15.0 Hz, 2H, H-25), 4.86, 4.75 (ABq, $J$ = 12.8 Hz, 2H, H-19), 4.61 (t, $J$ = 4.1 Hz, 1H, H-3), 4.31 (dd, $J$ = 6.9, 3.4 Hz, 1H, H-7), 4.06 (dd, $J$ = 8.6, 3.4 Hz, 1H, H-5), 3.96 (td, $J$ = 7.8, 6.6, 3.4 Hz, 3H, H-9 and H-11), 1.54 (s, 3H, –CH$_3$), 1.33 (s, 3H, –CH$_3$), 1.32 (s, 3H, –CH$_3$), 1.30 (s, 3H, –CH$_3$). $^{13}$C NMR (126 MHz, CDCl$_3$) $\delta$ 145.3, 134.9, 133.0, 129.4, 129.3, 122.9, 112.9, 109.6, 103.8, 79.90, 78.0, 77.8, 74.7, 65.0, 63.6 53.4, 26.8, 26.4, 26.1, 24.9. FT-IR/ATR $v_{max}$ cm$^{-1}$: 2991, 1476, 1369, 1261, 1206, 1106, 1012, 868, 786. HRMS (ESI-TOF) calculated for C$_{22}$H$_{29}$ClN$_3$O$_6$: [M + H]$^+$ 468.17154, found 468.17112.

### 4.4.9. 1,2:5,6-Di-*O*-isopropylidene-3-*O*-(1-(4-bromobenzyl)-1H-1,2,3-triazol-4-yl-methyl)-$\alpha$-D-allofuranose (**8d**)

White solid, 453 mg (89%), m.p. = 114–115°C, [$\alpha$]$_D$ + 63.0 (c 1, CHCl$_3$). $^1$H NMR (500 MHz, CDCl$_3$) $\delta$ 7.61 (s, 1H, H-21), 7.51 (d, $J$ = 8.4 Hz, 2H, ArH), 7.16 (d, $J$ = 8.4 Hz, 2H, ArH), 5.74 (d, $J$ = 3.6 Hz, 1H, H-2), 5.50, 5.45 (ABq, $J$ = 15.0 Hz, 2H, H-25), 4.86, 4.76 (ABq, $J$ = 12.1 Hz, 2H, H-19), 4.61 (t, $J$ = 4.0 Hz, 1H, H-3), 4.31 (td, $J$ = 6.8, 3.3 Hz, 1H, H-7), 4.06 (dd, $J$ = 8.6, 3.3 Hz, 1H, H-5), 4.04–3.87 (m, 3H, H-9 and H-11), 1.55 (s, 3H, –CH$_3$), 1.33 (s, 3H, –CH$_3$), 1.32 (s, 3H, –CH$_3$), 1.30 (s, 3H, –CH$_3$). $^{13}$C NMR (126 MHz, CDCl$_3$) $\delta$ 145.8, 133.5, 132.3, 129.7, 123.0, 112.9, 109.6, 103.8, 78.0, 78.0, 77.8, 74.7, 65.0, 63.6, 53.5, 26.8, 26.4, 26.1, 245.0. FT-

IR/ATR $v_{max}$ cm$^{-1}$: 2975, 1490, 1369, 1261, 1205, 1107, 1034, 868, 785. HRMS (ESI-TOF) calculated for C$_{22}$H$_{29}$BrN$_3$O$_6$ [M + H]$^+$ 510.12397, found 510.12319.

### 4.4.10. 1,2:5,6-Di-O-isopropylidene-3-O-(1-(4-iodobenzyl)-1H-1,2,3-triazol-4-yl-methyl)-α-D-allofuranose (8e)

White solid, 259 mg (53%), m.p. = 129–131°C; [α]$_D$ + 57.0. $^1$H NMR (500 MHz, CDCl$_3$) δ 7.71 (d, J = 8.4 Hz, 2H, ArH), 7.56 (s, 1H, H-21), 7.03 (d, J = 8.4 Hz, 2H, ArH), 5.74 (d, J = 3.8 Hz, 1H, H-2), 5.49, 5.43 (ABq, J = 15.0 Hz, 2H, H-25), 4.86, 4.76 (ABq, J = 12.8 Hz, 2H, H-19), 4.61 (t, J = 4.1 Hz, 1H, H-3), 4.31 (td, J = 6.8, 3.4 Hz, 1H, H-7), 4.06 (dd, J = 8.7, 3.4 Hz, 1H, H-5), 4.00–3.86 (m, 2H, H-9 and H-11), 1.54 (s, 3H, –CH$_3$), 1.33 (s, 3H, –CH$_3$), 1.31 (s, 3H, –CH$_3$), 1.30 (s, 3H, –CH$_3$). $^{13}$C NMR (126 MHz, CDCl$_3$) δ 145.4, 138.3, 134.0, 129.9, 129.9, 123.0, 112.9, 109.6, 103.8, 94.6, 78.2, 78.1, 78.00, 77.8, 74.8, 65.1, 53.6, 26.8, 26.5, 26.1, 25.0. FT-IR/ATR $v_{max}$ cm$^{-1}$: 2978, 1459, 1368, 1263, 1201, 1107, 1007, 868, 785. HRMS (ESI-TOF) calculated for C$_{22}$H$_{29}$IN$_3$O$_6$ [M + H]$^+$ 558.11010, found 558.06868.

## 4.5. General procedure for deprotection of carbohydrate-1,2,3-triazole conjugates

The corresponding glucosyl–triazol 7a–e or 8a–e (50 mg, 0.11 mmol) were dissolved in MeOH (4 ml) and then a solution of HCl (2 ml, 4 N) was added. The solution was stirred at room temperature for 5 h. After this time, the reaction mixture was treated with a saturated aqueous solution of NaHCO$_3$ until pH = 7 and extracted with ethyl acetate (3 × 30 ml). The organic phase was dried over Na$_2$SO$_4$ and evaporated under reduced pressure. The resultant residue was purified by column chromatography on silica gel.

### 4.5.1. 1,2-O-isopropylidene-3-O-(1-benzyl-1H-1,2,3-triazol-4-yl-methyl)-α-D-glucofuranose (9a)

Colourless syrup, 0.018 g (45%). $^1$H NMR (500 MHz, CDCl$_3$) δ 7.43–7.38 (m, 4H), 7.28–7.26 (m, 2H), 5.96 (d, J = 3.8 Hz, 1H), 5.54 (s, 2H), 5.19 (d, J = 4.6 Hz, 1H), 4.91 (d, J = 13.7 Hz, 1H), 4.65 (d, J = 13.7 Hz, 1H), 4.60 (d, J = 3.8 Hz, 1H), 4.17–4.11 (m, 2H), 4.10–4.07 (s, 1H), 3.87 (ddd, J = 10.7, 7.1, 3.2 Hz, 1H), 3.72–3.66 (m, 1H), 2.33 (br s, 1H), 1.47 (s, 3H), 1.33 (s, 3H). $^{13}$C NMR (126 MHz, CDCl$_3$) δ 144.6, 134.2, 129.3 (2C), 129.0, 128.3 (2C), 121.9, 111.9, 105.5, 82.5, 82.3, 80.43, 68.5, 64.7, 63.0, 54.5, 26.8, 26.3. HRMS (ESI-TOF) calculated for C$_{19}$H$_{26}$N$_3$O$_6$ [M + H]$^+$ 392.18216, found 392.18093.

### 4.5.2. 1,2-O-isopropylidene-3-O-(1-(4-fluorobenzyl)-1H-1,2,3-triazol-4-yl-methyl)-α-D-glucofuranose (9b)

Colourless syrup, 0.020 g (46%). $^1$H NMR (300 MHz, CDCl$_3$) δ 7.43 (s, 1H), 7.24 (dd, J = 8.8, 5.1 Hz, 2H), 7.03 (t, J = 8.6 Hz, 2H), 5.88 (d, J = 3.8 Hz, 1H), 5.44 (s, 2H), 5.03 (d, J = 4.6 Hz, 1H), 4.82 (d, J = 13.4 Hz, 1H), 4.62 (d, J = 13.3 Hz, 1H), 4.56 (d, J = 3.8 Hz, 1H), 4.13–4.02 (m, 2H), 4.02 (br s, 1H), 3.82 (br s, 1H), 3.63–3–62 (m, 1H), 2.89 (br s, 1H), 1.43 (s, 3H), 1.27 (s, 3H). $^{13}$C NMR (75 MHz, CDCl$_3$) δ 164.7, 161.4, 144.8, 130.37, 130.1, 121.9, 116.4, 116.1, 111.9, 105.5, 82.4, 82.2, 80.4, 68.5, 64.6, 63.01, 53.7, 26.8, 26.3. HRMS (ESI-TOF) calculated for C$_{19}$H$_{25}$FN$_3$O$_6$ [M + H]$^+$ 410.17274, found 410.18152.

### 4.5.3. 1,2-O-isopropylidene-3-O-(1-(4-chlorobenzyl)-1H-1,2,3-triazol-4-yl-methyl)-α-D-glucofuranose (9c)

Colourless syrup, 0.019 g (42%). $^1$H NMR (500 MHz, acetone-d$_6$) δ 8.04 (s, 1H), 7.47–7.35 (m, 4H), 5.84 (d, J = 3.8 Hz, 1H), 5.66 (s, 2H), 4.86 (dd, J = 13.0, 0.6 Hz, 1H), 4.77 (dd, J = 13.0, 0.6 Hz, 1H), 4.65 (d, J = 3.8 Hz, 1H), 4.50 (d, J = 5.3 Hz, 1H), 4.11–4.03 (m, 2H), 3.92 (dtd, J = 8.7, 5.6, 2.9 Hz, 1H), 3.75–3.71 (m, 1H), 3.58–3.49 (m, 2H), 1.41 (s, 3H), 1.27 (s, 3H). $^{13}$C NMR (126 MHz, acetone-d$_6$) δ 145.6, 135.6, 134.4, 130.5 (2C), 129.5 (2C), 123.82, 111.7, 106.0, 82.9, 82.5, 80.9, 69.3, 64.9, 63.7, 53.3, 26.9, 26.3. HRMS (ESI-TOF) calculated for C$_{19}$H$_{25}$ClN$_3$O$_6$ [M + H] + 426.14319, found 426.14884.

### 4.5.4. 1,2-O-isopropylidene-3-O-(1-(4-bromobenzyl)-1H-1,2,3-triazol-4-yl-methyl)-α-D-glucofuranose (9d)

Colourless syrup, 0.023 g (50%). $^1$H NMR (500 MHz, CDCl$_3$) δ 7.45 (d, J = 8.4 Hz, 2H), 7.34 (s, 1H), 7.09 (d, J = 8.4 Hz, 2H), 5.87 (d, J = 3.8 Hz, 1H), 5.40 (s, 2H), 4.81 (d, J = 13.6 Hz, 1H), 4.58 (d, J = 13.6 Hz, 1H), 4.52 (d, J = 3.8 Hz, 1H), 4.09–4.02 (m, 2H), 4.02–3.94 (m, 1H), 3.80 (dd, J = 11.3, 3.3 Hz, 1H), 3.62 (dd, J = 11.4, 6.1 Hz, 1H), 1.40 (s, 3H), 1.24 (s, 3H). $^{13}$C NMR (126 MHz, CDCl$_3$) δ144.8, 133.7, 133.2, 132.5 (2C), 129.9 (2C), 123.2, 111.9, 105.5, 82.4, 82.2, 80.5, 68.5, 64.7, 62.9, 53.8, 26.8, 26.3. HRMS (ESI-TOF) calculated for C$_{19}$H$_{25}$BrN$_3$O$_6$ [M + H]$^+$ 470.09267, found 470.09668.

### 4.5.5. 1,2-O-isopropylidene-3-O-(1-(4-iodobenzyl)-1H-1,2,3-triazol-4-yl-methyl)-α-D-glucofuranose (9e)

Colourless syrup, 0.018 g (40%). $^1$H NMR (500 MHz, acetone-d$_6$) δ 8.02 (s, 1H), 7.75 (d, $J$ = 8.5 Hz, 2H), 7.17 (d, $J$ = 8.6 Hz, 2H), 5.82 (d, $J$ = 3.8 Hz, 1H), 5.62 (s, 2H), 4.84 (dd, $J$ = 12.9, 0.6 Hz, 1H), 4.75 (dd, $J$ = 12.9, 0.6 Hz, 1H), 4.64 (d, $J$ = 3.8 Hz, 1H), 4.07–4.01 (m, 2H), 3.90 (ddd, $J$ = 9.0, 6.0, 3.2 Hz, 1H), 3.81 (s, 2H), 3.72–3.68 (m, 1H), 3.53 (dd, $J$ = 11.2, 6.1 Hz, 1H), 1.39 (s, 3H), 1.26 (s, 3H). $^{13}$C NMR (126 MHz, acetone-d$_6$) δ 145.7, 138.7 (2C), 136.7, 131.0 (2C), 124.0, 111.9, 106.2, 94.3, 83.0, 82.6, 81.1, 69.4, 64.8, 63.8, 53.7, 27.1, 26.5. HRMS (ESI-TOF) calculated for C$_{19}$H$_{25}$IN$_3$O$_6$ [M + H]$^+$ 518.07880, found 518.02245.

### 4.5.6. 1,2-O-isopropylidene-3-O-(1-benzyl-1H-1,2,3-triazol-4-yl-methyl)-α-D-allofuranose (10a)

Colourless syrup, 0.018 g (47%). $^1$H NMR (500 MHz, CDCl$_3$) δ 7.45 (s, 1H), 7.32–7.28 (m, 3H), 7.22–7.19 (m, 2H), 5.68 (d, $J$ = 3.7 Hz, 1H), 5.46 (d, $J$ = 22.3 Hz 1H), 5.44 (d, $J$ = 22.3 Hz, 1H), 4.80 (d, $J$ = 12.7 Hz, 1H), 4.65 (d, $J$ = 12.7 Hz, 1H), 4.60 (t, $J$ = 3.7 Hz, 1H), 3.99–3.87 (m, 2H), 3.88 (dd, $J$ = 8.4, 5.0 Hz, 1H), 3.56 (d, $J$ = 5.0 Hz, 2H), 3.05 (br s, 1H), 2.67 (br s, 1H), 1.56 (s, 3H), 1.32 (s, 3H). $^{13}$C NMR (126 MHz, acetone-d$_6$) δ 145.6, 136.9, 132.2, 129.7 (2C), 129.0 (2C), 128.8, 124.3, 112.9, 105.4, 80.1, 78.5, 78.5, 72.4, 63.8, 63.5, 54.3, 27.2, 27.0. HRMS (ESI-TOF) calculated for C$_{19}$H$_{26}$N$_3$O$_6$ [M + H]$^+$ 392.18216, found 392.18124.

### 4.5.7. 1,2-O-isopropylidene-3-O-(1-(4-fluorobenzyl)-1H-1,2,3-triazol-4-yl-methyl)-α-D-allofuranose (10b)

Colourless syrup, 0.018 g (38%). $^1$H NMR (500 MHz, acetone-d$_6$) δ 8.00 (s, 1H), 7.43 (dd, $J$ = 8.9, 5.3 Hz, 2H), 7.15 (t, $J$ = 8.9 Hz, 2H), 5.74 (d, $J$ = 3.8 Hz, 1H), 5.64 (s, 2H), 4.79 (dd, $J$ = 12.3, 0.6 Hz, 1H), 4.76 (t, $J$ = 4.0 Hz, 1H), 4.67 (dd, $J$ = 12.3, 0.6 Hz, 1H), 4.08–4.04 (m, 1H), 4.01 (dd, $J$ = 8.8, 3.2 Hz, 1H), 3.95 (d, $J$ = 4.2 Hz, 1H), 3.84–3.78 (m, 1H), 3.56–3.51 (m, 3H), 1.97 (s, 1H), 1.45 (s, 3H), 1.29 (s, 3H). $^{13}$C NMR (126 MHz, acetone-d6) δ 164.43, 145.70, 133.20, 131.11, 131.04, 124.30, 116.52, 116.35, 113.01, 105.36, 80.15, 78.56, 78.45, 72.62, 63.83, 63.69, 53.43, 27.20, 27.05. HRMS (ESI-TOF) calculated for C19H25FN3O6 [M + H]$^+$ 410.17274, found 410.18296.

### 4.5.8. 1,2-O-isopropylidene-3-O-(1-(4-chlorobenzyl)-1H-1,2,3-triazol-4-yl-methyl)-α-D-allofuranose (10c)

Colourless syrup, 0.020 g (43%). $^1$H NMR (400 MHz, CDCl$_3$) δ 7.55 (s, 1H), 7.36 (d, $J$ = 8.4 Hz, 2H), 7.22 (d, $J$ = 8.4 Hz, 2H), 5.76 (d, $J$ = 3.6 Hz, 1H), 5.52 (d, $J$ = 14.8 Hz, 1H), 5.50 (d, $J$ = 15.2 Hz, 1H), 4.88 (d, $J$ = 12.6 Hz, 1H), 4.72 (d, $J$ = 12.8 Hz, 1H), 4.69 (t, $J$ = 3.9 Hz, 1H), 4.03 (qd, $J$ = 8.9, 3.6 Hz, 2H), 3.96 (dd, $J$ = 8.4, 5.0 Hz, 1H), 3.66 (d, $J$ = 5.1 Hz, 2H), 3.20 (br s, 1H), 3.28 (br s, 1H), 1.56 (s, 3H), 1.33 (s, 3H). $^{13}$C NMR (101 MHz, CDCl$_3$) δ 144.6, 135.0, 133.0, 129.6 (2C), 129.5 (2C), 123.0, 113.3, 104.2, 79.0, 77.3, 77.16, 71.0, 63.2, 63.0, 53.66, 53.5, 26.88, 26.60. HRMS (ESI-TOF) calculated for C$_{19}$H$_{25}$ClN$_3$O$_6$ [M + H]$^+$ 426.14319, found 426.14412.

### 4.5.9. 1,2-O-isopropylidene-3-O-(1-(4-bromobenzyl)-1H-1,2,3-triazol-4-yl-methyl)-α-D-allofuranose (10d)

Colourless syrup, 0.018 g (42%). $^1$H NMR (300 MHz, CDCl$_3$) δ 7.54–7–50 (m, 3H), 7.15 (d, $J$ = 12 Hz, 2H), 5.76 (d, $J$ = 3.6 Hz, 1H), 5.57–5.34 (m, 2H), 4.88 (d, $J$ = 12.6 Hz, 1H), 4.75 (d, $J$ = 12.6 Hz, 1H), 4.07–3.99 (m, 2H), 3.96 (br s, 1H), 3.66 (t, $J$ = 5.1 Hz, 2H), 3.15 (br s, 1H), 2.79 (s, 1H), 2.79 (br s, 1H), 1.56 (s, 3H), 1.33 (s, 3H). $^{13}$C NMR (75 MHz, CDCl3) δ:144.66, 133.5, 132.5 (2C), 129.8 (2C), 123.19, 122.92, 113.31, 104.25, 78.99, 77.56, 77.37, 71.12, 63.23, 63.08, 53.71, 26.89, 26.62. HRMS (ESI-TOF) calculated for C$_{19}$H$_{25}$BrN$_3$O$_6$ [M + H]$^+$ 470.09267, found 470.09450.

### 4.5.10. 1,2-O-isopropylidene-3-O-(1-(4-iodobenzyl)-1H-1,2,3-triazol-4-yl-methyl)-α-D-allofuranose (10e)

Colourless syrup, 0.022 g (48%). $^1$H NMR (300 MHz, CDCl$_3$) δ 7.64 (d, $J$ = 8.5 Hz, 2H), 7.47 (s, 1H), 6.95 (d, $J$ = 8.5 Hz, 2H), 5.68 (d, $J$ = 3.7 Hz, 1H), 5.42 (d, $J$ = 20 Hz, 1H) 5.36 (d, $J$ = 20 Hz, 1H), 4.80 (d, $J$ = 12.9 Hz, 1H), 4.65 (d, $J$ = 12.7 Hz, 1H), 4.61 (t, $J$ = 3.7 Hz, 1H), 3.99–3.91 (m, 2H), 3.89 (br s, 1H), 3.58 (t, $J$ = 4.9 Hz, 2H), 3.12 (d, $J$ = 3.6 Hz, 1H), 2.76 (br s, 1H), 1.48 (s, 3H), 1.25 (s, 3H). $^{13}$C NMR (75 MHz, CDCl$_3$) δ 144.6, 138.4 (2C), 134.2, 130.0 (2C), 122.9, 113.3, 104.2, 94.8, 78.9, 77.5, 77.3, 71.1, 63.2, 63.0, 53.8, 26.9, 26.6. HRMS (ESI-TOF) calculated for C$_{19}$H$_{25}$IN$_3$O$_6$ [M + H]$^+$ 518.07880, found 518.02851.

## 4.6. Antimicrobial activity

The compounds **7a**–**e** were evaluated for their *in vitro* bactericidal and fungicidal action against Gram-positive bacteria, Gram-negative bacteria, yeasts and fungi. Penicillin, streptomycin and amphotericin B were used as standard antibiotics and antifungals for comparison with synthesized triazoles. The method used was disc diffusion (Kirby–Bauer test) recommended by the National Committee for Clinical Laboratory Standards (NCCLS) for the determination of bacterial sensitivity to antibiotics. The bacteria used were the following: *P. aeruginosa* and *E. coli* as Gram-negative bacteria; *S. aureus* and *B. subtilis* as Gram-positive bacteria; *C. utilis* as yeast; as well as the fungus *A. niger*. Gram-positive and negative bacteria together with yeasts were seeded in Luria-Bertani (LB) agar and Sabouraud dextrose agar was incubated for 24 h at a temperature of 37°C and 28°C, respectively. The fungus was seeded in papa dextrose agar and incubated for 7 days at 24°C. After this incubation, each bacterium, yeast and fungus was adjusted in LB broth to 0.5 of the McFarland tube. In parallel, a 300 µg ml$^{-1}$ solution of tested compounds was prepared. A massive striatum was performed in Petri dishes with Mueller–Hinton agar, each concentration of triazole and standard antibiotics was tested by triplicate, the bacteria and incubated at 37°C for 24 h, yeasts at 28°C for 18 h and the fungus was incubated for 7 days at 24°C. After that incubation time, the zone of inhibition formed in each Petri dish was measured in mm.

Data accessibility. The datasets supporting this article have been uploaded as part of the electronic supplementary material. Our data are deposited at the Dryad Digital Repository: https://datadryad.org/stash/share/UB3l0AFPAaFzddI_S0kJptVTu-tAROb1SDe6hjfQzu4 [46].

Authors' contributions. R.C.-S. and A.S.-E. carried out the synthesis, purification and characterization of organic compounds and wrote the manuscript. C.N.-L., Y.R.A. and A.C.R.-S. designed and performed the biological evaluation of synthesized compounds. G.E.N.-S. and L.L.-R. managed the research and assisted with designing, data analysis, writing and revising of the manuscript. All the authors gave their final approval for publication.

Competing interests. We declare we have no competing interests.

Funding. The authors thank CONACyT for the financial support granted for the development of this research through the project nos. 255819 and 1014. R.C.-S. wants to thank NPTC-PRODEP by SEP-Mexico for the financial support of this project.

Acknowledgements. R.C.-S., A.S.-E., L.L.-R. and G.E.N.-S. also wish to acknowledge the SNI for the distinction and the stipend received. The authors also thank Atilano Gutiérrez-Carrillo for support with the NMR spectroscopy.

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
