## [Reviewer comments · Royal Society Open Science]

Review History

RSOS-200290.R0 (Original submission)

Review form: Reviewer 1

Is the manuscript scientifically sound in its present form?

Yes

Are the interpretations and conclusions justified by the results?

Yes

Is the language acceptable?

Yes

Do you have any ethical concerns with this paper?

No

Have you any concerns about statistical analyses in this paper?

No

Recommendation?

Accept with minor revision (please list in comments)

Comments to the Author(s)

No special comments

Review form: Reviewer 2 (Charles Gauthier)**Is the manuscript scientifically sound in its present form?**

Yes

Are the interpretations and conclusions justified by the results?

Yes

Is the language acceptable?

Yes

Do you have any ethical concerns with this paper?

No

Have you any concerns about statistical analyses in this paper?

No

Recommendation?

Accept with minor revision (please list in comments)

Comments to the Author(s)

The submitted manuscript of Corona-Sanchez and co-workers deals with the synthesis of sugar-linked 1,2,3-triazole derivatives. The authors used a Cu-Al mixed oxide-sodium ascorbate system as a heterogeneous catalyst for the click reaction. Using their optimized protocol, the authors were able to synthesize an unprecedented series of glycoconjugates some of which showed interesting activities against microorganisms. The manuscript is well written, and compounds well characterized. I support the publication in Royal Society Open Science following the minor changes mentioned below:

Abstract, line 30: Change « synthetized » to « synthesized »

Abstract, line 38: IDEM

Page 2, line 48: Change "nitrogen containing" to "nitrogen-containing"

Figure 1: To be consistent, the pyranose of the two compounds at the bottom of Figure 1 should be drawn in their chair conformation.

Page 3, line 54: In the name of compound 1, the "O" must be italicized, and "D" written as a small capital letter. Please correct these mistakes anywhere they appear in the manuscript.

Scheme 1 caption: The "O" in "O-propargylated" must be italicized. Please add a "space" between "O" and "°C" (three times).

Scheme 1: Some of the compounds are known (if not all?). Therefore, for the known compounds it is not needed to add experimental details. Please start the Scheme with only the new transformations/compounds and add "ref. number" on the reaction arrows if needed.

Page 4, line 19: The "1" must be bolded.

Page 4, line 29: Please add a "space" between "O" and "°C". Please correct this anywhere it appears in the manuscript or in the SI.

Page 4, line 29: Delete the "space" between "con" and "figuration"

Page 4, line 30: The "4" must be bolded.

Page 4, line 41: Add "of" between "diffractogram" and "calcinated"

Page 5, line 40: Change “minutes” for “min”.

Page 5, line 41: Change “hours” for “h”.

Page 5, line 48: Delete the “space” between “83” and “%”. Please correct this anywhere it appears in the manuscript.

Table 3, Notes: Add a “space” between “4” and “N”. Change “hrs” to “h”.

Page 6, line 30: Change “synthetized” to “synthesized”

Page 9, line 25: Change “CDCl3” to CDCl₃ (3 must be subscripted)

Review form: Reviewer 3

Is the manuscript scientifically sound in its present form?

Yes

Are the interpretations and conclusions justified by the results?

Yes

Is the language acceptable?

Yes

Do you have any ethical concerns with this paper?

No

Have you any concerns about statistical analyses in this paper?

No

Recommendation?

Major revision is needed (please make suggestions in comments)

Comments to the Author(s)

In this manuscript, the authors report the synthesis of several carbohydrate-triazole conjugates, and their evaluation in disk-diffusion assays against pathogenic bacteria and fungi.

Carbohydrate-triazole conjugates have attracted considerable interest over the past few years. While many such examples have been reported, the specific examples in this manuscript appear to be new. Unusually, the key step in their synthesis, the 1,3-dipolar azide-alkyne cycloaddition reaction, is catalysed by a Cu-Al mixed oxide.

These two aspects of the work represent a sufficient advance of scientific knowledge to justify publication of the manuscript in Royal Society Open Science after appropriate revision.

MAJOR

(i) Surprisingly, there is no mention of the application of azido- or alkyne-sugars for metabolic glycoengineering. This is probably the most important and most common application of the azide-alkyne cycloaddition reaction in carbohydrate chemistry, pioneered by Bertozzi and others. This application should be briefly discussed in the introduction, and a relevant recent review cited (e.g., Carbohydrate Research 2016, 435, 121-141).

(ii) P4, line 32ff: “we have demonstrated that Cu-Al mixed oxide is capable of [...] giving exclusively the 1,4-disubstituted triazole” – what is the evidence for the formation of the 1,4-substituted triazole? (as opposed to the 1,5-substituted regioisomer) Please provide relevant evidence (e.g., from 1D and/or 2D NMR) and briefly discuss relevant diagnostic features.

(iii) P5, line 19: “the mixed oxide catalyst was [...] reused in the same click reaction with similar efficiency twice more” – please provide data to support this statement.

- (iv) Why were the synthetic intermediates 7a-e and 8a-e not deprotected FULLY? Please explain.
(v) Why were only the synthetic precursors of the glucofuranose series tested (i.e. 7a-e), and neither the corresponding allofuranose derivatives (8a-e), nor the final target molecules? (i.e. 9 and 10) Please explain.

MINOR

- (i) P2, line 18: The seminal publication by Meldal (Tornøe et al., J. Org. Chem., 2002, 67, 3057) on the Cu-catalysed azide-alkyne cycloaddition needs to be discussed (and cited!) alongside Sharpless' work.
(ii) P2, line 21: "due their hydroxyl groups can serve the purpose of place both the azide or the alkyne moieties needed for this click reaction." - This sentence is unclear, please rephrase.
(iii) P2, line 26: "hybrid 1,2,3-triazoles" - what is meant by "hybrid" triazoles?
(iv) P3, line 20: "has been informed" - has been reported
(v) P3, line 62: "10 mg the Cu-Al" - of the Cu-Al...
(vi) P13, line 57: Sthapylococcus aureus - typo

Decision letter (RSOS-200290.R0)

Dear Dr Negrón Silva:

Title: Cu-Al mixed oxide-catalyzed multicomponent synthesis of gluco- and allofuranose-linked 1,2,3-triazole derivatives
Manuscript ID: RSOS-200290

The editor assigned to your manuscript has now received comments from reviewers. We would like you to revise your paper in accordance with the referee and Subject Editor suggestions which can be found below (not including confidential reports to the Editor). Please note this decision does not guarantee eventual acceptance.

Please submit your revised paper before 21-Jun-2020. Please note that the revision deadline will expire at 00.00am on this date. If we do not hear from you within this time then it will be assumed that the paper has been withdrawn. In exceptional circumstances, extensions may be possible if agreed with the Editorial Office in advance. We do not allow multiple rounds of revision so we urge you to make every effort to fully address all of the comments at this stage. If deemed necessary by the Editors, your manuscript will be sent back to one or more of the original reviewers for assessment. If the original reviewers are not available we may invite new reviewers.

On behalf of the Subject Editor Professor Anthony Stace and the Associate Editor Dr Andrew Harned.

RSC Associate Editor:

Comments to the Author:

The referees have expressed enthusiasm for the work described in the manuscript. There are, however, several suggested revisions that should be addressed by the authors. Some are more involved than others, but all should be relatively straightforward to address.

RSC Subject Editor:

Comments to the Author:

(There are no comments.)

Reviewers' Comments to Author:

Reviewer: 1

Comments to the Author(s)

No special comments

Reviewer: 2

Comments to the Author(s)

The submitted manuscript of Corona-Sanchez and co-workers deals with the synthesis of sugar-linked 1,2,3-triazole derivatives. The authors used a Cu-Al mixed oxide-sodium ascorbate system as a heterogeneous catalyst for the click reaction. Using their optimized protocol, the authors were able to synthesize an unprecedented series of glycoconjugates some of which showed interesting activities against microorganisms. The manuscript is well written, and compounds well characterized. I support the publication in Royal Society Open Science following the minor changes mentioned below:

Abstract, line 30: Change « synthetized » to « synthesized»

Abstract, line 38: IDEM

Page 2, line 48: Change “nitrogen containing” to “nitrogen-containing”

Figure 1: To be consistent, the pyranose of the two compounds at the bottom of Figure 1 should be drawn in their chair conformation.

Page 3, line 54: In the name of compound 1, the “O” must be italicized, and “D” written as a small capital letter. Please correct these mistakes anywhere they appear in the manuscript.

Scheme 1 caption: The “O” in “O-propargylated” must be italicized. Please add a “space” between “O” and “^oC” (three times).

Scheme 1: Some of the compounds are known (if not all?). Therefore, for the known compounds it is not needed to add experimental details. Please start the Scheme with only the new transformations/compounds and add “ref. number” on the reaction arrows if needed.

Page 4, line 19: The “1” must be bolded.

Page 4, line 29: Please add a “space” between “O” and “^oC”. Please correct this anywhere it appears in the manuscript or in the SI.

Page 4, line 29: Delete the “space” between “con” and “figuration”

Page 4, line 30: The “4” must be bolded.

Page 4, line 41: Add “of” between “diffractogram” and “calcinated”

Page 5, line 40: Change “minutes” for “min”.

Page 5, line 41: Change “hours” for “h”.

Page 5, line 48: Delete the “space” between “83” and “%”. Please correct this anywhere it appears in the manuscript.

Table 3, Notes: Add a “space” between “4” and “N”. Change “hrs” to “h”.

Page 6, line 30: Change “synthetized” to “synthesized”

Page 9, line 25: Change “CDCl3” to CDCl₃” (3 must be subscripted)

Reviewer: 3

Comments to the Author(s)

In this manuscript, the authors report the synthesis of several carbohydrate-triazole conjugates, and their evaluation in disk-diffusion assays against pathogenic bacteria and fungi.

Carbohydrate-triazole conjugates have attracted considerable interest over the past few years. While many such examples have been reported, the specific examples in this manuscript appear to be new. Unusually, the key step in their synthesis, the 1,3-dipolar azide-alkyne cycloaddition reaction, is catalysed by a Cu-Al mixed oxide.

These two aspects of the work represent a sufficient advance of scientific knowledge to justify publication of the manuscript in Royal Society Open Science after appropriate revision.

MAJOR

(i) Surprisingly, there is no mention of the application of azido- or alkyne-sugars for metabolic glycoengineering. This is probably the most important and most common application of the azide-alkyne cycloaddition reaction in carbohydrate chemistry, pioneered by Bertozzi and others. This application should be briefly discussed in the introduction, and a relevant recent review cited (e.g., Carbohydrate Research 2016, 435, 121-141).

(ii) P4, line 32ff: “we have demonstrated that Cu-Al mixed oxide is capable of [...] giving exclusively the 1,4-disubstituted triazole” – what is the evidence for the formation of the 1,4-substituted triazole? (as opposed to the 1,5-substituted regioisomer) Please provide relevant evidence (e.g., from 1D and/or 2D NMR) and briefly discuss relevant diagnostic features.

(iii) P5, line 19: “the mixed oxide catalyst was [...] reused in the same click reaction with similar efficiency twice more” – please provide data to support this statement.

(iv) Why were the synthetic intermediates 7a-e and 8a-e not deprotected FULLY? Please explain.

(v) Why were only the synthetic precursors of the glucofuranose series tested (i.e. 7a-e), and neither the corresponding allofuranose derivatives (8a-e), nor the final target molecules? (i.e. 9 and 10) Please explain.

MINOR

(i) P2, line 18: The seminal publication by Meldal (Tornøe et al., J. Org. Chem., 2002, 67, 3057) on the Cu-catalysed azide-alkyne cycloaddition needs to be discussed (and cited!) alongside Sharpless' work.

(ii) P2, line 21: "due their hydroxyl groups can serve the purpose of place both the azide or the alkyne moieties needed for this click reaction." - This sentence is unclear, please rephrase.

(iii) P2, line 26: "hybrid 1,2,3-triazoles" - what is meant by "hybrid" triazoles?

(iv) P3, line 20: "has been informed" - has been reported

(v) P3, line 62: "10 mg the Cu-Al" - of the Cu-Al...

(vi) P13, line 57: Sthapylococcus aureus - typo

Author's Response to Decision Letter for (RSOS-200290.R0)

See Appendix A.

Decision letter (RSOS-200290.R1)

Dear Dr Negrón Silva:

Title: Cu-Al mixed oxide-catalyzed multicomponent synthesis of gluco- and allofuranose-linked 1,2,3-triazole derivatives

Manuscript ID: RSOS-200290.R1

It is a pleasure to accept your manuscript in its current form for publication in Royal Society Open Science. The chemistry content of Royal Society Open Science is published in collaboration with the Royal Society of Chemistry.

Royal Society of Chemistry
Thomas Graham House
Science Park, Milton Road

Cambridge, CB4 0WF
Royal Society Open Science - Chemistry Editorial Office

On behalf of the Subject Editor Professor Anthony Stace and the Associate Editor Dr Andrew Harned.

RSC Associate Editor

Comments to the Author:

The authors have done an excellent job at responding to the previous review. I believe this manuscript is acceptable in its current form.

Reviewer(s)' Comments to Author:

Appendix A

Universidad Autónoma Metropolitana-Azcapotzalco
Laboratorio de Química de Materiales
Departamento de Ciencias Básicas
División de Ciencias Básicas e Ingeniería
June 2020

Dr. Laura Smith
Editor of Royal Society Open Science

Dear Dr Smith

Attached to this letter, you will find a revised version of our manuscript entitled “Cu-Al mixed oxide-catalyzed multicomponent synthesis of gluco- and allofuranose-linked 1,2,3-triazole derivatives” (RSOS-200290). All the suggestions and recommendations of the reviewers have been considered. A point-by-point explanation of the changes made into the manuscript is provided below.

Comments:

Reviewer #1

- *Table 1 8e, write 8^e*

This typo was corrected

- *Preparation of the catalyst (p4, line 36): I quote you: previously reported method, from $\text{CuNO}_3 \cdot 2.5\text{H}_2\text{O}$ (Write: $\text{Cu}(\text{NO}_3)_2$) and $\text{AlCl}_3 \cdot 6\text{H}_2\text{O}$ by the co-precipitation with Na_2CO_3 and NaOH in water.[18] In the literature 18 it was reported that you use $\text{Cu}(\text{NO}_3)_2$ and $\text{Al}(\text{NO}_3)_3$ but no NaOH which method is correct?.*

We thank the reviewer for highlighting this observation. The correct method for the preparation of the catalyst is the one described in reference 18. In our group we synthesized several hydrotalcites by co-precipitation method and the preparation of some of them involves the use of NaOH , however this is not true for the Cu-Al hydrotalcite. The paragraph was rewritten as follows:

Firstly, catalyst was prepared, using our previously reported method, from $\text{Cu}(\text{NO}_3)_2 \cdot 2.5\text{H}_2\text{O}$ and $\text{AlCl}_3 \cdot 6\text{H}_2\text{O}$ by co-precipitation with Na_2CO_3 in water.[18]

- *P 5 line 52: and it is necessary the presence of an external stabilizing ligand / write: and need to add an external stabilizing ligand.*

The sentence was modified as suggested by the reviewer.

- *x-becomes in a drawback / Write: becomes a drawback*

The sentence was modified as suggested by the reviewer.

- *P 5 line 53: Write: Thus, we demonstrate that the described method is efficient and don't need any additional stabilizing ligands.*

The sentence was modified as suggested by the reviewer.

- *P 6 line 6: Write: to perform this reaction.*

The sentence was modified as suggested by the reviewer.

Reviewer #2

- *Abstract, line 30: Change « synthesized » to « synthesized »*

This typo was corrected

- *Abstract, line 38: IDEM*

This typo was corrected

- *Page 2, line 48: Change “nitrogen containing” to “nitrogen-containing”*

This typo was corrected

- *Figure 1: To be consistent, the pyranose of the two compounds at the bottom of Figure 1 should be drawn in their chair conformation.*

We thank the reviewer for highlighting this observation. The figure 1 was modified as suggested by the reviewer as follows:

Figure 1 Some triazolyl-glycoconjugates with biological properties.

- *Page 3, line 54: In the name of compound 1, the “O” must be italicized, and “D” written as a small capital letter. Please correct these mistakes anywhere they appear in the manuscript.*

These typos were corrected along all the manuscript.

- *Scheme 1 caption: The “O” in “O-propargylated” must be italicized. Please add a “space” between “0” and “°C” (three times).*

The “O” was italicized in the caption. There was no need to correct the other typos since the reaction conditions were removed from the scheme by suggestion of reviewer.

- *Scheme 1: Some of the compounds are known (if not all?). Therefore, for the known compounds it is not needed to add experimental details. Please start the Scheme with only the new transformations/compounds and add “ref. number” on the reaction arrows if needed.*

We agree with the reviewer. The Scheme 1 was modified as suggested by the reviewer as follows:

Scheme 1. Synthesis of *O*-propargylated glucose diacetone derivatives.

- *Page 4, line 19: The “1” must be bolded.*

This typo was corrected

- *Page 4, line 29: Please add a “space” between “0” and “°C”. Please correct this anywhere it appears in the manuscript or in the SI.*

These typos were corrected along all the manuscript and supporting information.

- *Page 4, line 29: Delete the “space” between “con” and “figuration”*

This typo was corrected

- *Page 4, line 30: The “4” must be bolded.*

This typo was corrected

- *Page 4, line 41: Add “of” between “diffractogram” and “calcinated”*

This typo was corrected

- *Page 5, line 40: Change “minutes” for “min”.*

The word was changed as suggested by the reviewer.

- *Page 5, line 41: Change “hours” for “h”.*

This word was changed as suggested by the reviewer.

- *Page 5, line 48: Delete the “space” between “83” and “%”. Please correct this anywhere it appears in the manuscript.*

This typo was corrected

- *Table 3, Notes: Add a “space” between “4” and “N”. Change “hrs” to “h”.*

This typo was corrected

- *Page 6, line 30: Change “synthetized” to “synthesized”*

This typo was corrected

- *Page 9, line 25: Change “CDCl3” to CDCl₃” (3 must be subscripted)*

These typos were corrected

Reviewer #3

- *Surprisingly, there is no mention of the application of azido- or alkyne-sugars for metabolic glycoengineering. This is probably the most important and most common application of the azide-alkyne cycloaddition reaction in carbohydrate chemistry, pioneered by Bertozzi and others. This application should be briefly discussed in the introduction, and a relevant recent review cited (e.g., Carbohydrate Research 2016, 435, 121-141).*

We agree with the reviewer. In the introduction of the manuscript we included a brief discussion about this topic and cited the recommended literature accordingly. The following paragraph was added to the manuscript:

Click chemistry has great potential for use in binding between nucleic acids, lipids, proteins, and carbohydrates. In this context, metabolic glycoengineering allows for the direct modification of living cells with substrates for click chemistry either *in vitro* or *in vivo* becoming in a powerful tool for cell transplantation and drug delivery.[16]

16. Sminia, T. J., Zuilhof, H., Wennekes, T. 2016. Getting a grip on glycans: A current overview of the metabolic oligosaccharide engineering toolbox. *Carbohydr. Res.*, **435**, 121-141. (doi.org/10.1016/j.carres.2016.09.007)

- *P4, line 32ff: “we have demonstrated that Cu-Al mixed oxide is capable of [...] giving exclusively the 1,4-disubstituted triazole” – what is the evidence for the formation of the 1,4-substituted triazole? (as opposed to the 1,5-substituted regioisomer) Please provide relevant evidence (e.g., from 1D and/or 2D NMR) and briefly discuss relevant diagnostic features.*

We agree with the reviewer. The regioselectivity of the cycloaddition reaction was elucidated by using ^1H - ^{13}C long-range NMR experiments (HMBC). The following paragraph was added to the manuscript as well as the figures shown below:

The regioselectivity of the cycloaddition reaction was elucidated by using ^1H - ^{13}C long-range NMR experiments (HMBC). The selective formation of 1,4-disubstituted 1,2,3-triazoles was confirmed by a long-range correlation between the methylene protons H-19 and H-25 with the carbon bearing H-21 in the triazole ring since in 1,5-regioisomer the correlation between the benzyl protons and the carbon bearing the proton in the triazole ring would not be present (Figure 5).

Figure 5 Expansion of ^1H - ^{13}C HMBC spectrum highlighting the correlations of triazole ring in compound **7c**

All the figures were renumbered in captions and text according to this new numeration.

- *P5, line 19: “the mixed oxide catalyst was [...] reused in the same click reaction with similar efficiency twice more” – please provide data to support this statement.*

We thank the reviewer for highlighting this observation. For support such statement we included following figure in the manuscript:

Figure 3 Recyclability of catalyst for the three-component 1,3-dipolar azide-alkyne cycloaddition

- *Why were the synthetic intermediates 7a-e and 8a-e not deprotected FULLY? Please explain.*

It is well known in carbohydrate chemistry that substrates having multiple acetonide protections might be selectively hydrolyzed in the less hindered terminal isopropylidene ketal. The use of a Brønsted acid is usually selective for the less hindered terminal isopropylidene ketal deprotection if the reaction time is carefully controlled. In this work, we are interested in this selective deprotection since in our group we are currently using the free primary alcohol to functionalize this kind of derivatives without the inherent problems to work with secondary alcohols.

- *Why were only the synthetic precursors of the glucofuranose series tested (i.e. 7a-e), and neither the corresponding allofuranose derivatives (8a-e), nor the final target molecules? (i.e. 9 and 10) Please explain.*

As indicated in the title, the main goal of this work is to develop a new method for the synthesis of carbohydrate linked-1,2,3-triazole derivatives using as key step a three-component 1,3-dipolar azide-alkyne cycloaddition catalyzed by Cu-Al mixed oxide. As a part of preliminary studies, biological activity of glucofuranose series derivatives were tested as antimicrobial compounds. Due to the results of biological activity, which indicated that these triazoles show only low to moderate antimicrobial activity against the tested strains, we decided not to continue the study with the diastereomeric series of allofuranose derivatives because we expected that these compounds exhibit similar biological properties.

- *P2, line 18: The seminal publication by Meldal (Tornøe et al., J. Org. Chem., 2002, 67, 3057) on the Cu-catalysed azide-alkyne cycloaddition needs to be discussed (and cited!) alongside Sharpless' work.*

We agree with the reviewer. We mentioned and cited the work of Medal group in the introduction section. The following paragraph was added to the manuscript:

Compounds containing the 1,2,3-triazole ring have gained increased attention in the drug-discovery field since the introduction of the "click" chemistry concept which was reported simultaneously and independently by the groups of Meldal in Denmark[13] and Fokin and Sharpless in the U.S. [14]

All the references were renumbered in both manuscript and references section accordingly.

P2, line 21: “due their hydroxyl groups can serve the purpose of place both the azide or the alkyne moieties needed for this click reaction.” – This sentence is unclear, please rephrase.

We agree with the reviewer. The sentence was rephrased as follow:

In this sense, carbohydrates represent versatile starting substrates for the CuAAC reaction due is possible to synthesize from them both the azide- or the terminal alkyne-containing moieties needed for this cycloaddition reaction.

- *P2, line 26: “hybrid 1,2,3-triazoles” – what is meant by “hybrid” triazoles?*

Hybrid molecules are defined as chemical entities with two or more structural domains having different biological functions and dual activity, indicating that a hybrid molecule acts as two distinct pharmacophores. Frequently, these two pharmacophores are linked through a triazole ring and are called “hybrid 1,2,3-triazoles”.

- *P3, line 20: “has been informed” – has been reported*

We agree with the reviewer. The verb “informed” was replaced by “reported”.

- *P3, line 62: “10 mg the Cu-Al” – of the Cu-Al...*

This typo was corrected

- *P13, line 57: *Sthapylococcus aureus* - typo*

This typo was corrected

We hope that these modifications will address the concerns of the reviewers and that the manuscript can be accepted for publication in Royal Society Open Science. Moreover, we would like to thank the reviewers for their interesting suggestions that have contributed to improving the manuscript.

Sincerely

Dr. Guillermo Negrón Silva